

# Towards suitable description of reference architectures

Pedro Henrique Dias Valle[1], Lina Garcés[2], Tiago Volpato[1], Silverio Martínez-Fernández[3] and Elisa Yumi Nakagawa[1]

[1] University of São Paulo (USP), São Carlos, Brazil
[2] Federal University of Itajubá (UNIFEI), Itajubá, Brazil
[3] Universitat Politècnica de Catalunya (UPC-BarcelonaTech), Barcelona, Spain

## ABSTRACT

Due to the increasing size and complexity of many current software systems, the architectural design of these systems has become a considerably complicated task. In this scenario, reference architectures have already proven to be very relevant to support the architectural design of systems in diverse critical application domains, such as health, avionics, transportation, and the automotive sector. However, these architectures are described in many different approaches, such as using textual description, informal models, and even modeling languages as UML. Hence, practitioners are faced with a difficult decision of the better approaches to describing reference architectures. The main contribution of this work is to depict a detailed panorama containing the state of the art (from the literature) and state of the practice (based on existing reference architectures) of approaches for describing reference architectures. For this, we firstly examined the existing approaches (e.g., processes, methods, models, and modeling languages) and compared them concerning completeness and applicability. We also examined four well-known, successful reference architectures (AUTOSAR, ARC-IT, IIRA, and AXMEDIS) in view of the approaches used to describe them. As a result, there exists a misalignment between the state of the art and state of the practice, requiring an engagement of the software architecture community, through research collaboration of academia and industry, to propose more suitable means to describe reference architectures and, as a consequence, promoting the sustainability of these architectures.

# INTRODUCTION

Software systems have continually increased in size and complexity and, as a consequence, the design of their architecture has become a critical issue (*Garlan, 2000*). Besides that, software architectures play a fundamental role in determining the system's quality, as they are responsible for addressing quality characteristics, such as interoperability, performance, portability, adaptability, and maintainability (*Bass, 2013*). According to *Bass (2013)*, software architecture is the structure or structures of the system composed of software components, the externally visible properties of those components, and the relationships among them. In this scenario, many reference architectures have emerged

Corresponding authors
Pedro Henrique Dias Valle,
pedrohenriquevalle@usp.br
Elisa Yumi Nakagawa,
elisa@icmc.usp.br

as a solution to support the development of critical software-intensive systems in the industry (*Galster et al., 2017*; *Nakagawa et al., 2015*). A reference architecture refers to architecture at a higher level of abstraction compared with the architecture of given software systems. It aggregates knowledge about how to design software architectures of systems of a given application domain (*Bass, 2013*; *Nakagawa, Oliveira & Becker, 2011*). It includes domain business rules, standards and legislation, software and hardware elements, architectural styles and patterns, and best practices of software development in that domain, among other elements (*Angelov, Grefen & Greefhorst, 2012*; *Martínez-Fernández et al., 2014*; *Nakagawa, Oliveira & Becker, 2011*). Hence, the main purpose of reference architectures is to serve as a guide for the development, standardization, and evolution of systems (*Nakagawa et al., 2014*; *Yimam & Fernandez, 2016*; *Martínez-Fernández et al., 2017*). Diverse application domains have already been benefited from reference architectures, such as the automotive sector (*AUTOSAR, 2020*), ambient assisted living (*Bayer et al., 2004*), big data systems (*Sang, Xu & Vrieze, 2016*), smart cities (*Schieferdecker et al., 2017*), and Industry 4.0 (*Industrial Internet Consortium, 2020*).

From the industry perspective, *Martnez-Fernandez et al. (2015)* identified benefits of reference architectures: (i) systematic reuse of common functionalities and configurations throughout the development of systems; (ii) risk reduction through the use of proven and partly qualified architectural elements included in the reference architecture; (iii) enhanced quality by facilitating the achievement of software quality aspects already addressed by the reference architecture; and (iv) interoperability among different systems and their software components establishing common means for information exchange. However, to obtain such benefits, these architectures should be suitably described (i.e., represented/modeled) aiming at reliably communicating the knowledge that they contained.

The description of software architectures is mainly used to improve the communication and cooperation among stakeholders, enabling them to work in an integrated, coherent way during the development and evolution of software systems (*International Organization for Standardization, 2011*). Such descriptions are tangible artifacts that contain relevant information about the systems and are also commonly used to evaluate alternative architectures and as input for simulation tools (*International Organization for Standardization, 2011*). In particular, for reference architectures, we can observe that their descriptions are found in diverse formats and containing different elements, making sometimes difficult the comprehension and, as a consequence, the dissemination of these architectures. Besides that, practitioners are also faced with a difficult decision to choose suitable approaches for describing reference architectures. To the best of our knowledge, there is not still a wider investigation on the existing approaches to describe reference architectures and even which ones could contribute to making these architectures sustainable, i.e., architectures with the capacity to endure different types of changes through efficient maintenance and orderly evolution over their entire life cycle (*Avgeriou, Stal & Hilliard, 2013*).

Motivated by this scenario, the main contribution of this work is to present a detailed panorama of the approaches (e.g., processes, methods, models, and architecture description languages—ADL) or describing reference architectures. Such panorama

depicts both the state of the art (collected from the literature) and the state of the practice (observed from the existing reference architectures). For this, we identified 19 approaches that were deeply examined regarding their completeness and applicability. Following, we analyzed four well-known, large, and successful reference architectures (namely, AUTOSAR (*AUTOSAR, 2020*), ARC-IT (*USA, 2019*), IIRA (*Industrial Internet Consortium, 2020*), and AXMEDIS (*Bellini & Nesi, 2005*) to get the state of the practice and understand how they were described. These architectures are widely disseminated and used in industry and academia and are supported by large communities and/or consortia that involve various companies and/or research institutions or universities; besides, they have a long-term existence focusing on very important domains for the society and, more importantly, they present a good documentation that has been continuously updated. We also analyzed elements contained in these four architectures that could be contributing to some extent to making them sustainable over time. As a result, we observe a large distance between the state of the art and the state of the practice. While the state of the art encompasses approaches presented in a higher level of abstraction, without real-world evaluations and, more importantly, without fully considering the international standard for architecture description (i.e., ISO/IEC/IEEE 42010 (*International Organization for Standardization, 2011*)), the state of the  practice encompasses particular approaches that have worked well in the reference architectures and, to some extent, have made these architectures sustainable. Besides that, there is a lack of generic approaches that explicitly concern the sustainability of reference architectures.

This work is organized as follows. 'Background and Related Work' presents background and related work. 'Research Method' presents the research method, while 'Results' discusses results, including the analysis of the four reference architectures. Following, 'Discussion' discusses the main findings and threats to the validity of this work. Finally, 'Final Remarks' presents the final remarks.

# BACKGROUND AND RELATED WORK

This section brings an overview of reference architectures, software architecture description,[1] and sustainability of reference architectures. Following, it presents the related work.

## Reference architectures

During the last around 30 years, both academia and industry have invested effort to consolidate the area of reference architecture by proposing definitions to reference architectures (*Kruchten, 2000*; *Nakagawa, Oliveira & Becker, 2011*), their benefits and drawbacks (*Martínez-Fernández et al., 2017*), and means to engineer (*Angelov, Grefen & Greefhorst, 2012*; *Nakagawa et al., 2014*; *Muller, 2008*; *Galster & Avgeriou, 2011*) and describe them (*Eklund et al., 2012*; *Guessi, Oquendo & Nakagawa, 2014b*; *Gherardi & Brugali, 2014*).

Reference architectures can be used to provide (*Muller, 2008*): (i) a common lexicon and taxonomy that facilitate the communication among stakeholders; (ii) a common architectural vision, which manages the efforts of the several people and teams involved;

[1] In the context of this work, architectural *description*, *representation*, and *modeling* are used as synonymous.

and (iii) modularization and complementary context that assist in the division and integration of efforts posteriorly. It is worth highlighting that, more importantly, reference architectures avoid the reinvention and revalidation of solutions to problems that were already solved (*Nakagawa, Oliveira & Becker, 2011*).

To systematize the building of reference architectures, the scientific community has already contributed with different initiatives. *Muller (2008)* proposed recommendations to build and evolve reference architectures, where these architectures should be easy to understand and evolve. *Bayer et al. (2004)* and *Pohl, Böckle & Linden (2005)* proposed a systematic approach to define reference architectures from the knowledge of existing systems in the context of software product line (SPL). *Cloutier et al. (2010)* presented a high-level model for reference architecture development in systems engineering. *Nakagawa et al. (2014)* proposed a process, called ProSA-RA, that systematizes the design, representation, and evaluation of reference architectures. *Angelov, Grefen & Greefhorst (2012)* developed a classification that can support the design of reference architectures. Finally, *Galster & Avgeriou (2011)* proposed a six-step procedure for reference architecture design. It is important to observe these different approaches include an activity for architectural description of reference architectures, but without detailing or specifying guidelines for that. Hence, other complementary studies, like those found in this work and discussed in 'Results' have emerged to cover this lack.

## Software architecture description

Serving as an important support to the communication and cooperation in software project teams, the architecture description of a software system should be adequately available to a variety of stakeholders. An architecture description should serve as (*International Organization for Standardization, 2011*): (i) a baseline for system design and development activities; (ii) a baseline to analyze and evaluate alternative implementations of an architecture; (iii) a support to the system development and maintenance; (iv) a support to document characteristics, features, and design of a system for potential clients, acquirers, owners, operators, and integrators; (v) a basis to analyze and evaluate alternative architectures; and (vi) a means to share lessons learned and reuse architectural knowledge through viewpoints, patterns, and styles.

The ISO/IEC/IEEE 42010 established definitions and relationships among the main elements that compose architecture descriptions, e.g., stakeholder, concern, architecture decision, architecture view, architecture viewpoint, and architecture model, but it does not suggest or impose any specific process, method, model, notation, or technique to produce an architecture description. Hence, this standard can serve as a basis for different approaches, such as document-centric, model-based, and repository-based techniques (*International Organization for Standardization, 2011*). Due to this flexibility, this standard becomes popular and is to some extent widely adopted by both academia and industry.

With regard to views to describe software architectures, *Kruchten (1995)* proposed 4+1 view model containing five views: (i) *logic view* that shows the components (objects) of the system and their interactions; (ii) *process view* that shows processes/workflow rules of a system and how these processes communicate with each other; (iii) *development view*

that presents a building block view of the system; (iv) *physical view* that shows the system execution environment; and (v) *scenario view* (also use case view) that shows a set of use cases serving to illustrate and validate the architecture design. Another well-established work is "*Views and Beyond*" by *Clements et al. (2011)* and, to describe an architecture, most relevant architectural views are firstly documented, and then additional documentation to the views are developed. Views are classified into three main categories: (i) *modular view* that describes the structure of the system as a set of implementation units; (ii) *component-and-connector view* that describes the structure of the system at the time it is running; and (iii) *implementation view* that describes how the system relates to other structures in its environment.

## Sustainability of reference architectures

Sustainability was brought to the software architecture area as an important concept related to the capacity of software architectures to tolerate modifications throughout the software systems life cycle (*Avgeriou, Stal & Hilliard, 2013*). In parallel, due to reference architectures encompass a valuable knowledge of a given domain, their sustainability is also considered of utmost importance.

While several reference architectures have been proposed for various application domains, many of them have not survived. For instance, *Volpato et al. (2017)* analyzed 20 reference architectures, most of them destined to software systems based on service-oriented architecture (SOA), an architectural style widely adopted to develop software-intensive systems for different and even critical domains. Results showed 12 of them did not present any evidence (publications, projects, and/or websites) indicating updates or initiatives for using or disseminating them. In addition, these architectures did not have a good architectural description in the sense that it provided good support for the use and dissemination of these architectures. It is important to mention that other factors, such as financial support, economic viability, and the existence of a consortium, also impact the sustainability of reference architectures (*Volpato et al., 2017*).

On the other hand, reference architectures that have a good description have survived for decades, being constantly updated accordingly to the advance of their application domain. For instance, AUTOSAR, a well-known reference architecture for the automotive sector, adopts an update policy with release and version control of its documentation to manage evolution (*Venters et al., 2018*). Their current version is described in 22,271 pages organized into 220 files. The same occurs in other reference architectures, such as AXMEDIS and ARC-IT with life cycles of over 14 and 25 years, respectively.

In this scenario, sustainability in the context of reference architectures can have two perspectives (*Volpato et al., 2017*): (i) the perspective "IN" is about understanding how sustainable the concrete software architectures that are instantiated from a given reference architecture are; and (ii) the perspective "OF" (which is addressed in this work) refers to how the reference architectures themselves are sustainable. Regarding this last perspective, this study also highlights the reference architecture description must be continually updated and aligned with the state of practice to achieve sustainable architectures; also, this study

exemplifies eight reference architectures that have sustained over time by keeping their description updated.

### Related work

With regard to the related work, we identified a systematic literature review (SLR) on architectural description of software architectures and reference architectures of embedded systems (*Guessi et al., 2012*). This work identified 24 studies to answer: (i) how software architectures and reference architectures of the embedded systems have been modeled; and (ii) which approaches have been adopted for that. As the main result, the authors concluded that there is no consensus on how to better describe the architectures of embedded systems. They also identified a range of quality requirements and constraints that have been considered in the architectural description of embedded systems.

Another SLR was conducted to understand how Systems-of-Systems (SoS) software architectures have been described (*Guessi et al., 2015*). The authors selected 38 primary studies to answer their research questions: (i) how the literature has addressed the architecture description of SoS; and (ii) which techniques have been used in the description of software architectures of SoS. The authors suggested that more research should be conducted for effectively using architecture descriptions in the evaluation and evolution of SoS. They also proposed a set of research lines to be further addressed, including the establishment of architecture viewpoints framing important quality attributes for SoS and a consensus on the formalism level required at each stage of their life cycle.

As far as we know, there are not literature surveys, systematic mapping study (SMS), or other SLR on approaches for describing reference architectures. Then, the novelty of our work is to present a wider panorama of these approaches (independently of the domain of the reference architectures or type of systems) and also analyze how well-known, successful reference architectures have been described.

## RESEARCH METHOD

To support the definition of the panorama of the approaches to describe reference architectures, we conducted an SMS and also examined four well-known, large, and successful reference architectures. The planning and conduction of the research method are presented in 'Planning', 'Conduction' respectively.

### Planning

We adopted the GQM (*Goal Question Metric*) approach (*Basili, Caldiera & Rombach, 1994*) to support the conduction of our SMS and also to examine the four reference architectures. GQM is composed of three parts (*Basili, Caldiera & Rombach, 1994*): (i) the *goal* to be achieved; (ii) a set of *questions* that must be answered to achieve the goal; and (iii) a set of *metrics* needed to answer the questions. Hence, the goal of this work is:

**Analyze** approaches to describe reference architectures

**for the purpose of** their evaluation and classification

**with respect to** adherence to the ISO/IEC/IEEE 42010

**from the viewpoint of the** software engineering research

**Table 1  Research questions and metrics.**

| Research Questions | Metrics |
| --- | --- |
| $RQ_1$: Which approaches have been proposed to describe reference architectures? | $M_{1.1}$: Approaches proposed by year |
| | $M_{1.2}$: Approaches proposed in academia and industry contexts |
| | $M_{1.3}$: Approaches proposed for reference architectures in specific applications domains |
| | $M_{1.4}$: Types of contribution (i.e., process, frameworks, methods, models) |
| $RQ_2$: Which is the adherence level of approaches to describe reference architectures to the standard ISO/IEC/IEEE 42010? | $M_{2.1}$: Types of architectural views considered (if so) in the approach |
| | $M_{2.2}$: Types of architectural viewpoints considered (if so) in the approach |
| | $M_{2.3}$: Types of models considered in the approach |
| | $M_{2.4}$: Class of stakeholders defined in the approach |
| | $M_{2.5}$: Concerns types described in the approach |
| | $M_{2.6}$: Architectural decisions considered in the approach |
| | $M_{2.7}$: Rationale description strategies by the approach |
| | $M_{2.8}$: ADL proposed by the approach |
| | $M_{2.9}$: Types of architectural decisions (i.e., architectural patterns, styles, technologies) used by the approach |
| $RQ_3$: How sustainable reference architectures have been described? | $M_{3.1}$: Year of establishment |
| | $M_{3.2}$: Number of pages in the first and the last version |
| | $M_{3.3}$: Dissemination of reference architecture |
| | $M_{3.4}$: Life cycle |
| | $M_{3.5}$: Number of releases |
| | $M_{3.6}$: ISO/IEC/IEEE 42010 Adherence Level |
| | $M_{3.7}$: Description of approaches |

[2]We used the version of 2011 of ISO/IEC/IEEE 42010 in this work since the last version (of 2020) has just been submitted to the ISO Secretariat for balloting.

**in the context of** sustainability.

It is worth highlighting that we adopted the international standard ISO/IEC/IEEE 42010.[2]

Table 1 presents the three research questions (RQs) and their respective metrics.

RQ1 aims to collect possibly all existing approaches to describe reference architectures through metrics $M_{1.1}$ to $M_{1.4}$. RQ2 intends to analyze the adherence of the approaches to ISO/IEC/IEEE 42010, which is an international standard for architecture descriptions of systems and software. Metrics $M_{2.1}$ to $M_{2.9}$ aim to collect nine elements directly related to the description of reference architectures: view, viewpoint, type of model, stakeholder, concern, architectural decision, rationale, ADL, and type of architectural decisions. RQ3

**Table 2  Search strings adapted for each publication database.**

| Source | Search string |
| --- | --- |
| Scopus | TITLE-ABS-KEY((''software architecture'') AND (''software architecture'' OR ''software structure'' OR ''software design'' OR ''system architecture'' OR ''system structure'' OR ''system design'') ) |
| IEEE Xplore | (''Abstract'':reference architecture) AND (''Abstract'':software architecture OR ''Abstract'':software structure OR ''Abstract'':software design OR ''Abstract'':system architecture OR ''Abstract'':system structure OR ''Abstract'':system design) |
| ACM Library | [[[Abstract: ''reference architecture''] AND [[Abstract: ''software architecture''] OR [Abstract: ''software structure''] OR [Abstract: ''software design''] OR [Abstract: ''software structure''] OR [Abstract: ''system structure''] OR [Abstract: ''system design'']]] |
| Web of Science | AB=((''software architecture'') AND (''software architecture'' OR ''software structure'' OR ''software design'' OR ''system architecture'' OR ''system structure'' OR ''system design'') ) |
| ScienceDirect | Title, abstract, keywords: ((''software architecture'') AND (''software architecture'' OR ''software structure'' OR ''software design'' OR ''system architecture'' OR ''system structure'' OR ''system design'') ) |
| SpringerLink | ((''software architecture'') AND (''software architecture'' OR ''software structure'' OR ''software design'' OR ''system architecture'' OR ''system structure'' OR ''system design'') ) |

aims to analyze how well-known, successful reference architectures have been described and is answered by examining four reference architectures through metrics $M_{3.1}$ to $M_{3.7}$.

To define the search string of our SMS, we selected two keywords: *reference architecture* and *software architecture*. As *reference architecture* is a well-known, disseminated term, we did not consider other related terms. Otherwise, we considered the following similar terms of *software architecture* as also used in *Qureshi, Usman & Ikram (2013)*: software structure, software design, system architecture, system structure, and system design. Hence, the final search string[3] was: *(''reference architecture'' and (''software architecture'' or ''software structure'' or ''software design'' or ''system architecture'' or ''system structure'' or ''system design'')).* As shown in Table 2, we accordingly adapted this string to the specific syntax of each publication database to perform the searches.

With regard to the publication databases, we selected those recommended in (*Kitchenham et al. (2009)*: Scopus (http://scopus.com), Web of Science (http://isiknowledge.com), IEEE Xplore (https://ieeexplore.ieee.org), ACM Digital Library (http://dl.acm.org), ScienceDirect (http://sciencedirect.com), and SpringerLink (http://link.springer.com). Scopus, ScienceDirect, and Web of Science are general indexing systems and allow us to cover a broader scope for our search. IEEE Xplore, ACM Digital Library, and SpringerLink publish works of the most important venues (conferences and journals) related to software

[3]For this search string, we also consider the plural form of all terms, but for simplification, only the singular terms are showed here.

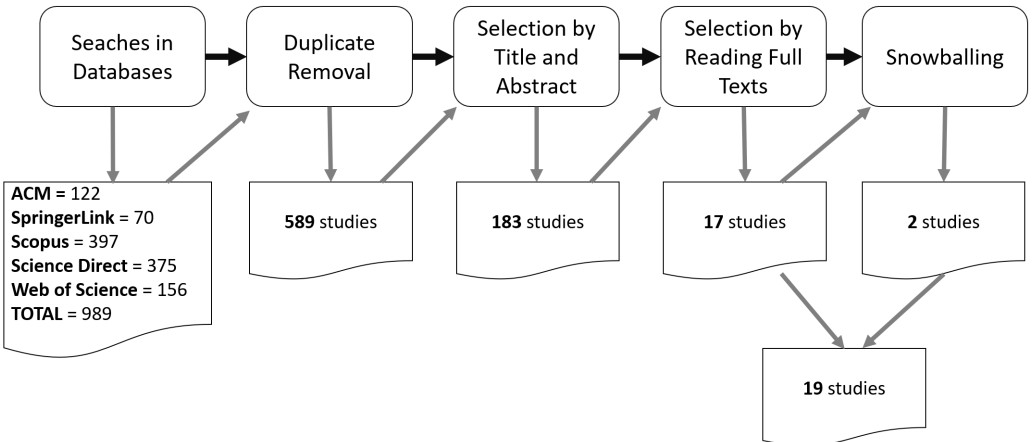

**Figure 1** **Process for the selection of relevant primary studies.** .

architectures. We also defined one inclusion criterion (IC) and three exclusion criteria (EC):

1. IC1: The study proposes an approach to describe reference architectures.
2. EC1: The study does not address an approach to describe reference architectures.
3. EC2: The study does not permit to identify information about the approaches, because it is a table of contents, short course description, invited talk of events, summary of events, among others, or written in other languages than English.
4. EC3: The study was not peer-reviewed.

## Conduction

This SMS was first conducted from January to July 2018 by four researchers from both industry and academia and with experience in reference architectures and software architectures, besides their experience in researching, conducting, and updating a number of SMS and SLR. Figure 1 depicts the steps of the selection process. By adapting the search string for each database, we performed the searches and obtained a total of 989 studies. After removing the duplicated studies, 589 studies remained. After the first selection where we applied the selection criteria on title, abstract, and keywords, 183 studies were selected. After reading the full text of these studies and applying the selection criteria again, 17 studies were finally selected. Besides that, an inspection of the list of references of each selected study (through snowballing approach (*Wohlin, 2014*)) made us possible to include other two relevant studies, totaling 19 studies. To support this selection process, we used JabRef (http://www.jabref.org).

Considering the need of updating our SMS (*Mendes et al., 2020*), in December 2020, we updated it aiming at including studies published in 2018 (i.e., months not covered in the first version of the SMS), 2019, and 2020. For that, we used the snowballing approach that refers to using the reference list of a study or the citations to a given study to identify additional studies. According to *Wohlin (2014)*, snowballing is particularly useful for extending an SLR or SMS, since new studies almost certainly must cite at least one study among the

previously relevant studies. Hence, snowballing is by deduction a better approach than a database search for extending SMS (*Wohlin, 2014*). *Felizardo et al. (2016)* also found that snowballing presents an overall precision to search for new studies similar to searches in databases.

In particular, we applied forward snowballing, since it could identify new studies published in 2018, 2019, and 2020. For that, the reference lists of the 19 studies initially included in our SMS were considered. Following, we used information made available in Google Scholar to find all studies that cited those 19 studies and, as a result, we identified 108 candidate studies. We checked the title, abstract, and keywords of the 108 studies and applied the inclusion and exclusion criteria. Hence, 82 studies were excluded and 26 remained. Following, we read the full text of the 26 studies and applied again the inclusion and exclusion criteria. As a result, no new study published between 2018 and 2020 was suitable to answer our RQ; hence, no new study was included in our SMS.

We used an online form for the data extraction and analysis of each study. This form was designed to collect data to answer RQ1 and RQ2. Data from each study was then extracted by one researcher involved in this study and when there were doubts, discussions with other researchers were conducted. The dataset gathered from this form together with a qualitative and qualitative analysis supported us to synthesize results, answer these RQs, and further draw conclusions.

To answer RQ3 and identify the state of the practice about how reference architecture has been described, we examined four reference architectures (AUTOSAR, ARC-IT, IIRA, and AXMEDIS), which are from different application domains. Based on our experience of more than 15 years at researching and establishing reference architectures, we selected such architectures because they are widely known in the industry, besides presenting long-time existence. We also have previously investigated them in our research group, i.e., we have followed the evolution of these architectures over the years.

## RESULTS

Section 'Overview of studies' firstly presents an overview of the 19 studies resulting from the SMS, while 'Approaches to reference architecture description', 'Adherence of the Approaches to ISO/IEC/IEEE 42010', 'Analysis of four successful reference architectures' answer, respectively, RQ1, RQ2, and RQ3.

### Overview of studies

Table 3 lists the 19 studies included in our SMS, together with their ID (S1 to S19), title, publication year, reference, publication venue (i.e., W = workshop, TR = technical report, C = conference, J = journal, or BC = book chapter), context where the approach was developed (i.e., A = academia or I = industry), quantity of reference architectures described using the approach, domain for which the approach was created, and type of the approach (e.g., process, method, model, among others).

It is observed that the first three studies (S1, S2, and S3, published in 1994, 1998, and 1998, respectively) were published in workshops when the first events in this area were proposed. Hence, regarding the publication venues, while around one-third of studies were

Dias Valle et al. (2021), *PeerJ Comput. Sci.*, DOI 10.7717/peerj-cs.392

**Table 3** Approaches to describe reference architectures (Venue: C = Conference, W = Workshop, J = Journal, TR = Technical Report, BC = Book chapter); (Context: A = Academia, I = Industry)).

| ID | Title | Year | Ref. | Venue | Context | RA | Domain | Type |
|----|-------|------|------|-------|---------|-----|--------|------|
| S1 | A reference architecture for control of mechanical systems | 1994 | *Kramer et al. (1994)* | W | A | 1 | Mechanical systems | Process |
| S2 | NSA's MISSI reference architecture - Moving from prose to precise specifications | 1998 | *Meldal & Luckham (1998)* | W | A | 0 | Generic | ADL |
| S3 | PuLSE-DSSAa method for the development of software reference architectures | 1998 | *DeBaud, Flege & Knauber (1998)* | W | I | 0 | Generic | Method |
| S4 | Describing, instantiating and evaluating a reference architecture: A case study | 2003 | *Avgeriou (2003)* | TR | A | 0 | Generic | Method |
| S5 | Definition of reference architectures based on existing systems | 2004 | *Bayer et al. (2004)* | TR | I | 0 | Generic | Process |
| S6 | An Approach to Reference Architecture Design for Different Domains of Embedded Systems | 2008 | *Dobrica & Niemelä (2008)* | C | A | 0 | Generic | Method |
| S7 | Architectural Knowledge in an SOA Infrastructure Reference Architecture | 2009 | *Zimmermann, Kopp & Pappe (2009)* | BC | I | 0 | Generic | Method |
| S8 | A Methodology for Developing an Agent Systems Reference Architecture | 2011 | *Nguyen et al. (2011)* | W | A | 0 | Generic | Process |
| S9 | A reference architecture for integrated EHR in Colombia | 2011 | *Cruz et al. (2011)* | J | A | 0 | Agent Systems | Process |
| S10 | Empirically-grounded reference architectures: A proposal | 2011 | *Galster & Avgeriou (2011)* | C | A | 1 | Health | Process |
| S11 | A reference architecture template for software-intensive embedded systems | 2012 | *Eklund et al. (2012)* | C | A | 0 | Generic | Document template |
| S12 | RAModel: A Reference Model for Reference Architectures | 2012 | *Nakagawa, Oquendo & Becker (2012)* | C | A | 0 | Generic | Model |
| S13 | Towards a bottom-up development of reference architectures for smart energy systems | 2013 | *Irlbeck et al. (2013)* | W | I | 0 | Smart Energy Systems | Process |
| S14 | An approach for capturing and documenting architectural decisions of reference architectures | 2014 | *Guessi, Oquendo & Nakagawa (2014a)* | C | A | 0 | Generic | Method |
| S15 | Development and Specification of a Reference Architecture for Agent-Based Systems | 2014 | *Regli et al. (2014)* | J | A | 0 | Agent Systems | Process |

Dias Valle et al. (2021), *PeerJ Comput. Sci.*, DOI 10.7717/peerj-cs.392

**Table 3** (*continued*)

| ID | Title | Year | Ref. | Venue | Context | RA | Domain | Type |
|----|-------|------|------|-------|---------|----|--------|------|
| S16 | Modeling and reusing robotic software architectures: The HyperFlex toolchain | 2014 | *Gherardi & Brugali (2014)* | C | A | 1 | Robotic | Process |
| S17 | Variability viewpoint to describe reference architectures | 2014 | *Guessi, Oquendo & Nakagawa (2014b)* | C | A | 0 | Generic | Viewpoint |
| S18 | Design and Evaluation of a Customizable Multi-domain Reference Architecture on Top of Product Lines of Self-driving Heavy Vehicles: An Industrial Case Study | 2015 | *Schroeder et al. (2015)* | C | A | 1 | Automotive | Process |
| S19 | Quality-based heuristic for optimal product derivation in Software Product Lines | 2015 | *Losavio & Ordaz (2015)* | C | A | 1 | Generic | Process |

published in workshops, around half part of the studies were published in conferences; besides that, only two studies were published in journals, equally as the two technical reports, and only one book chapter. The concentration of studies in events (conference and workshops) may be related to the fact that studies are not still enough mature to be published in high-impact journals.

It is also worth highlighting that the first studies were published in 1994 and 1998 and, after that, there is a gap until 2003 when other studies started to be published. This gap may have happened because only after the 2000s, reference architectures started to show their value to the software systems development and, hence, they became more popular together with the software architecture area. Consequently, studies concerning representation/description of these architectures become necessary and have been published in the last almost two decades.

Moreover, around one-third of the studies focused on describing reference architectures of specific application domains, mainly those critical, while around two-thirds intended to be generic enough to address different domains. In these 19 studies, we found the description of only five reference architectures using the proposed approaches. Similarly, only four studies were developed in the industry context. Hence, the research topic of reference architecture description is still relatively new and requires to be matured and disseminated.

## Approaches to reference architecture description

This section deeps the analysis of the existing approaches through data collected using metrics $M_{1.1}$ (Number of approaches proposed by year), $M_{1.2}$ (Number of approaches proposed in academia and industry, i.e., the Context), $M_{1.3}$ (Application Domains targeted by the approach), and $M_{1.4}$ (Type of contribution). Regarding $M_{1.1}$, the first study was published in the early of 1990s and most of them are concentrated in the last decade, as illustrated in Fig. 2, showing a trend to an increased interest in this area. Concerning the other three metrics, a summary of the analysis is shown in Fig. 3 and is detailed below.

### Context of the approaches

To find the context of the approaches, for each study, we analyzed all authors' affiliations and how the development and evaluation of such approaches were performed. As result, we found four studies (S3, S5, S7, and S13) were carried out in the industry. In particular, S3 and S5 were authored by Fraunhofer Institute for Experimental Software Engineering Fraunhofer IESE) (https://www.iese.fraunhofer.de/), in Germany, a leading research institution in the area of software and systems engineering, while S7 was authored by IBM Research and IBM Global Technology Services, located in Switzerland and Germany, respectively. Regarding S13, one of its authors is part of Fortiss GmbH, located in Germany, a research institution for software-intensive systems and services. **Finding 1**: *All contributions from industry had the participation of German industry.*

The remaining 15 studies were proposed and validated in the academic context from different institutions in North and Latin America, Europe, and Asia, more specifically, the United States, Germany, France, Netherlands, Sweden, Switzerland, Finland, Cyprus, Romania, Brazil, Colombia, Venezuela, and China.

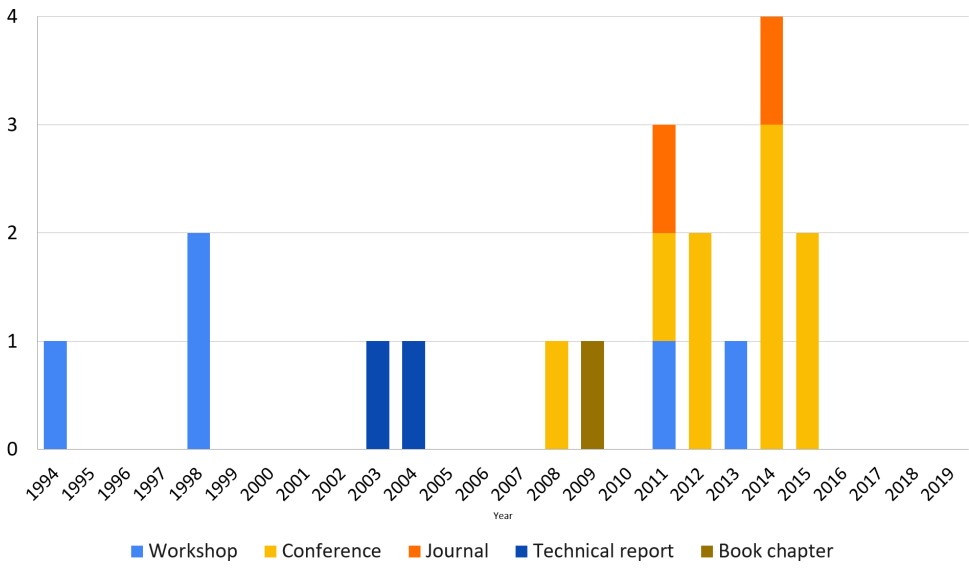

**Figure 2** Amount of studies by year and publication venue.

### Domains targeted by the approaches

We checked whether the approaches were proposed for specific domains or for general purpose. As summarized in Fig. 3, seven studies considered a particular application domain. Two of them (S9 and S15) focused on describing reference architectures for agent-based systems, while the other studies (S1, S10, S13, S16, and S18) considered, respectively, mechanical, health, smart energy, robotics, and automotive systems. Most of these studies were proposed in the academic context. The remaining 12 studies targeted a general purpose solution, i.e., they presented means that could be used to describe any reference architectures independently of their application domain. It is worth highlighting that three of four studies that had involvement of industry also aimed generic purpose solutions. **Finding 2:** *Most approaches are generic and could serve to describe reference architectures independently from their domain. However, generic approaches are overall less detailed than approaches for specific domains, as expected.*

### Types of contribution

From the studies, we identified six different types of contributions (i.e., process, method, ADL, reference model, architecture viewpoint, and architectural template), as presented in Fig. 3.

Most studies proposed processes to support the description of reference architectures. A **process** can be defined as a logical sequence of tasks performed to achieve a particular objective. It defines *what* is to be done, without specifying *how* each task is performed. We identified 10 processes, as listed in Table 4, including two studies (S5 and S13) conducted in the industry. While S5 presented a process to describe reference architectures from experience accumulated of existing systems, S13 presented a process for the incremental description of reference architectures for smart energy systems.

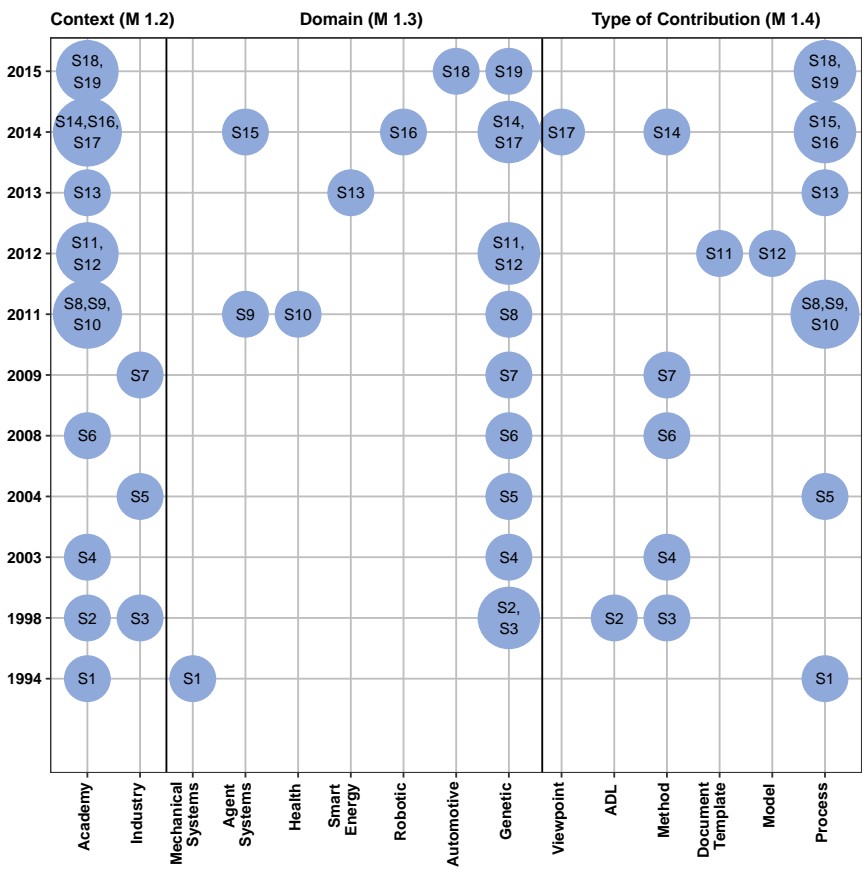

**Figure 3** Characterizing approaches for reference architecture description with regard to the Context (measured using $M_{1.2}$), Domain ($M_{1.3}$), and Type of contribution ($M_{1.4}$).

**Table 4 Processes for describing reference architectures.**

| ID | Architectural design activities | Context | Evaluation |
|---|---|---|---|
| S1 | Analysis and Synthesis | Academy | No Evaluation |
| S5 | Synthesis | Industry | Case study |
| S8 | Synthesis | Academy | Case study |
| S9 | Analysis and Synthesis | Academy | No Evaluation |
| S10 | Analysis and Synthesis | Academy | No Evaluation |
| S13 | Analysis, Synthesis, and Evaluation | Academy | Case study |
| S15 | Analysis and Synthesis | Academy | Case study |
| S16 | Analysis and Synthesis | Academy | Case study |
| S18 | Analysis and Synthesis | Academy | Case study |
| S19 | Synthesis | Academy | No Evaluation |

We also analyzed the coverage of each process comparing them with the Holfmeister et al.'s generic architectural process (*Hofmeister et al., 2007*), which presents three main activities: analysis, synthesis, and evaluation. While some studies encompassed the

architectural analysis (that addresses requirements of reference architectures), all of them considered the synthesis, in which the reference architecture description itself is performed. Differently from other studies, S13 considered all three activities, including a means to evaluate reference architectures. Regarding the maturity of the processes, in general, an effective evaluation is still widely missing. In particular, only S5 was evaluated in the real-world industry scenarios and, therefore, they could be considered more mature than the others. Otherwise, S1, S9, S10, and S19 only presented the steps contained in the processes without any evaluation.

Studies also provided methods for describing reference architecture. In the context of this work, a **method** refers to a means to perform a task, i.e., the *how* of that task. We classified the identified approaches as a method when they also used the terms technique, practice, and procedure and identified five studies (S3, S4, S6, S7, and S14). S3 and S7 were carried out in the industry context. While S3 proposed the systematic, iterative method to describe reference architectures for SPL, S7 showed an industrial case study to create and use architectural knowledge to describe reference architectures. For this, the authors introduced knowledge about the business domain, service portfolio, and knowledge management. S4, S6, and S14 carried out case studies to evaluate the applicability of the methods proposed by them. S4 presented an architecture instance that was designed for the development of a prototype of a learning management system. In S6, the authors presented an example using their method to model a reference architecture for embedded systems. The main contribution of this study was the synthesis of the most important issues of product-line architectures in their development strategy for cross-domain architecture design of systems-of-systems. In S14, the authors illustrated a method for documenting architectural decisions into a reference architecture design process.

We identified only one study (S2) that discussed the use of a formal ADL to model reference architectures. An **ADL** is any form of expression that can be used to the architecture descriptions (*International Organization for Standardization, 2011*). It provides one or more model kinds as a means to frame some concerns for the audience of stakeholders. In S2, the authors discussed the reading of an architecture description, mainly about the question of what the description actually means needs to be resolved unambiguously in the readers' and designers' minds to evaluate and then implement a given architecture. In particular, this ADL (in this case, *Rapide*) presents an event-based architecture model, i.e., the architecture components are defined by the kinds of events that they may generate or react to. In short, the authors concluded that Rapide allowed drawing unambiguous conclusions from the formalization based on testable arguments. As a contribution to the reference architecture area, Rapide can provide architects with the opportunity to define architectures in a descriptive rather than a prescriptive manner. Besides that, it is important to highlight that semi-formal languages were also found in the studies. For instance, S6 used UML-RT, a real-time extension of UML, to express the architecture views of the reference architecture. Both S5 and S13 from the industry also suggested UML to model the views and viewpoints of reference architectures.

Concerning the variety of elements that different reference architectures contained, S12 presented a **reference model**, called RAModel, which outlines the elements that should be

contained in reference architectures (*Nakagawa, Oquendo & Becker, 2012*). This model also aimed to improve the understanding of what reference architectures are and, therefore, it intended to support the design, use, and evolution of such architectures.

There are different architecture viewpoints and views used to represent reference architectures, as further detailed in 'Analysis of four successful reference architectures'. However, we found a proposal (S17) to describe variability in reference architectures. Such variability is not usually found in the description of existing reference architectures. S17 proposed an architecture viewpoint, the steps to create this viewpoint, and a technique to represent it. In turn, an **architecture viewpoint** refers to a representation of one or more aspects of an architecture that illustrates how it addresses the concerns held by one or more of its stakeholders (*International Organization for Standardization, 2011*).

We also found a document template for describing reference architectures. A **document template** addresses the somewhat conflicting needs when documenting a reference architecture. S11 presented a document template that prescribes two separate documents: (i) one document captures essential principles and evolution of the reference architecture; and (ii) another document captures technical details, providing the foundation for the implementation of concrete architecture. Besides, this template makes it possible to document and manage subsequent versions/releases of the reference architecture description.

Overall, the 19 approaches identified in this SMS were presented at a higher level of abstraction and without detailed guides that can support architects to easily apply them. **Finding 3**: *Contributions from different perspectives from processes to document template for describing reference architectures have been proposed, but they should mature in the sense they become more widely experimented with and used in academic and mainly industry context.*

### Adherence of the approaches to ISO/IEC/IEEE 42010

Figure 4 summarizes the results.

An **architecture view** considers one or more of the concerns held by the system's stakeholders, i.e., it expresses the architecture of a system from the perspective of specific system concerns (*International Organization for Standardization, 2011*). To identify the architecture views in the studies, we considered those views proposed, used, and/or cited throughout each study. Table 5 shows the 21 different views (exactly as mentioned and found in nine studies), together with a description of each view.

An **architecture viewpoint** establishes the conventions for the construction, interpretation, and use of architecture views to frame specific system concerns (*International Organization for Standardization, 2011*). We identified 25 architectural viewpoints that were proposed, used, and/or cited in one-third of the studies (6 of 19). Table 6 presents these viewpoints and studies that addressed them. Still in this table, we present a description of each viewpoint to help architects and researchers to better understand them and further select them to represent their reference architectures. As shown in Table 6, S5 is the most complete study compared with the others by proposing seven viewpoints, while S17 only considered one viewpoint.

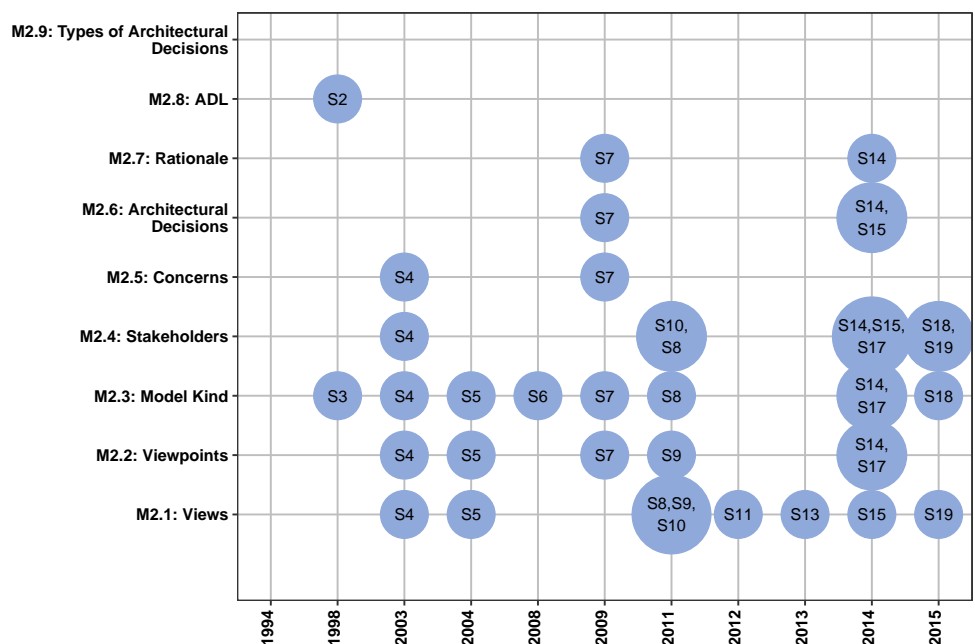

**Figure 4** Adherence of the approaches to the international standard ISO/IEC/IEEE 42010.

A **model kind** defines the conventions for one type of architecture model (*International Organization for Standardization, 2011*). We identified 13 different model kinds (i.e., diagrams) to describe reference architectures in approximately half part of the studies (9 of 19), as shown in Table 7. Model kinds that can represent the behavior of components and systems built from the reference architectures were the most recurrent in the studies. More specifically, S3, S4, S5, S7, and S8 used UML behavior diagrams: use case diagram, activity diagram, sequence diagram, state diagram, and collaboration diagram. UML structure diagrams were also suggested: component diagram, package diagram, and class diagram. Moreover, S17 explored the SysML internal block diagram, while two studies (S3 and S5) used the workflow diagram.

A **stakeholder** can be an individual, team, or organization that have an interest in a system (*International Organization for Standardization, 2011*). As presented in Table 8, we identified ten different stakeholders in six studies to be considered during the description of reference architectures, including mainly software architects, project managers, and developers. However, these studies did not present *how* to involve them and *which* their tasks are.

A **concern** refers to any interest in the system (*International Organization for Standardization, 2011*). A concern can appear in different forms, such as quality attributes, architecture decisions, risks, and other issues. Only two studies (S4 and S7) addressed concerns, but few details were presented. S4 provided means to describe the stakeholders' concerns in the viewpoints and, for this, a set of questions guide architects to understand stakeholders' concerns. S7 considered concerns related to business rules to describe

**Table 5  Architectural views addressed by the approaches for describing reference architectures.**

| Views | Studies | | | | | | | | | Description |
|---|---|---|---|---|---|---|---|---|---|---|
| | S4 | S5 | S8 | S9 | S10 | S11 | S13 | S15 | S19 | |
| Functional Logical | ✓ | ✓ | | | ✓ | | | ✓ | ✓ | It describes the most important classes, their organization in packages and subsystems, and the organization of these packages and subsystems into layers. |
| Process | | ✓ | ✓ | ✓ | | | | ✓ | | It describes the design concurrency and synchronization aspects. |
| Components | | ✓ | | ✓ | | | ✓ | | | It shows the components and topologies needed for the development of an instance of the system family or for the development of the domain. |
| Implementation | | ✓ | ✓ | ✓ | | | | ✓ | | It describes the package layout of the system from the perspective of the system architect. |
| Scenario | | ✓ | ✓ | | | | | ✓ | | This crosscutting view is composed of narrative use cases to provide an executive-level view of the architecture. |
| Platform | | ✓ | | ✓ | | | | | | It shows the elements (including hardware, operating systems, and middleware), their topology, and the allocation of software components to hardware. |
| Technical | | ✓ | | ✓ | | | | | | It defines the components and might refine components of the logical view. This view is used when detailed architectures are needed |
| Physical | | ✓ | | | | | | | | It describes the mapping of the software onto the hardware and reflects its distributed aspects. |
| Context | | ✓ | | | | | | | | It shows dynamic system properties such as capacity, liveness, and correct behavior, and all the ilities of a system such as reliability and maintainability. |
| Informal | | | | | | ✓ | | | | It describes both how to logically solve the upgrade problem and what components need to be active. |
| Information Models | | | | ✓ | | | | | | It is used to describe the data required. This is accomplished through the use of schemes, which describe the state and structures. |
| Domain | | ✓ | | | | | | | | It shows the problem space and what functions and capabilities must be provided, which are common and which are variable across a family, and how the functions are interrelated through information flow or in cooperation to provide capabilities. |
| Interface | | ✓ | | | | | | | | It is architectural views as a means of communication vehicle between design and recovery, and among stakeholders. |
| Code | | ✓ | | | | | | | | It isolates the construction and development aspects of a software system, and organize them in a separate view according to the organizations particular development environment. |
| Module | | ✓ | | | | | | | | It organizes modules into two orthogonal structures: decomposition and layers. The decomposition of a system captures the way a system is decomposed into a hierarchy of subsystems and modules. |

 

**Table 5** (*continued*)

| Views | Studies | | | | | | | | | Description |
|---|---|---|---|---|---|---|---|---|---|---|
| | **S4** | **S5** | **S8** | **S9** | **S10** | **S11** | **S13** | **S15** | **S19** | |
| Execution | | ✓ | | | | | | | | It comprises the runtime aspects of the software system and explains the deployment of the system and how the elements of the code, module, and conceptual view can be mapped to concrete external elements. |
| Conceptual | | ✓ | | | | | | | | This view is closest to the application domain. It can be a key facilitator to interact with domain experts who are not interested in the details of the software system, but in what the system does in terms of domain concepts. |
| New | | ✓ | | | | | | | | It creates a new representation for the elements and relationships defined in the meta-model. |
| Filtered | | ✓ | | | | | | | | It filters out elements in an existing view (in case they are not important for the new view) or highlighted (in case they are the focus of attention). An example of highlighting is a structural architecture view in which the elements that are made persistent are marked for a persistence view. |
| Augmented | | ✓ | | | | | | | | It adds new elements to an existing view, for example, annotations for performance data in dynamic views. |
| Deployment | ✓ | | | | | | | | | It concerns the identification of the various computational nodes and protocols specified in the reference architecture. In other words, it depicts all the system servers that are connected to the application server through appropriate protocols. |

reference architectures, but few details were provided on *how* these concerns should be considered.

An **architecture decision** affects the architecture description elements and pertains to one or more concerns (*International Organization for Standardization, 2011*). Only three studies (S7, S14, and S15) addressed architectural decisions for the design of reference architectures. S7 represented architectural decisions in a semi-formal way using architectural patterns and a meta-model, while S14 and S15 represented such decisions in an informal way through text description.

A **rationale** refers to the explanation, justification, or reasoning about architecture decisions that have been made and also architectural alternatives not chosen (*International Organization for Standardization, 2011*). Only S7 and S14 considered rationale. S7 addressed rationales for architectural decision-making through a textual description, while S14 used a meta-model and textual description to represent rationales.

An **ADL** refers to any form of expression for the architecture description (*International Organization for Standardization, 2011*). Some representative examples are Rapide, SysML, and ArchiMate. Only S2 considered a formal ADL, namely Rapide ADL, for the reference

**Table 6  Architectural viewpoints addressed by the approaches for describing reference architectures.**

| Viewpoint | Studies | | | | | | Description |
|---|---|---|---|---|---|---|---|
| | S4 | S5 | S7 | S9 | S14 | S17 | |
| Use-case | ✓ | | | | | | It describes a certain behavior of the system by capturing how the static elements of the conceptual architecture view or the static modules of the module view interact in order to show the activities and the order in which a scenario is realized |
| Logical | ✓ | | | | | | It shows the decomposition and behavior of the system at a logical level of abstraction |
| Deployment | ✓ | | | | | | It shows how one or more applications are realized on the infrastructure |
| Implementation | ✓ | | | | | | It is concerned with the technical representation of a system and the technologies and system components required for implementing the activities and functions prescribed |
| Data | ✓ | | | | | | It shows the persistent data that are stored and manipulated by the system |
| Build-time architecture | | ✓ | | | | | It can close the gap between the code architecture and the execution architecture view by explicitly describing the build process and its elements |
| Behavioral | | ✓ | | | | | It captures how the structural elements of a software system interact for given scenarios |
| Execution | | ✓ | | | | | It comprises the runtime aspects of the software system and explains the deployment of the system and how the elements of the code, module, and the conceptual view can be mapped into concrete external elements. |
| Code architecture | | ✓ | | | | | It isolates the construction and development aspects of a software system, and organize them in a separate view according to the organizations particular development environment |
| Module architecture | | ✓ | | | | | It organizes modules into two orthogonal structures: decomposition and layers. In the module view, all the application functionality, control functionality, adaptation, and mediation must be mapped to the module |
| Conceptual architecture | | ✓ | | | | | It describes the method used to extract conceptual components from User manuals |
| Feature | | ✓ | | | | | It shows parts of the feature model (features and some relationships) can be found in the documentation |
| Physical | | | ✓ | | | | It represents physical elements that operate in the field and the back-office, the functionality contained within those elements, the roles elements play in delivering user services, and the connections between those elements |
| Scenario | | | ✓ | | | | It describes the architecture using a small set of use cases, or scenarios, which become a fifth view. The scenarios describe sequences of interactions between objects and between processes. They are used to identify architectural elements and to illustrate and validate the architectural design |

**Table 6** (*continued*)

| Viewpoint | Studies | | | | | | Description |
|---|---|---|---|---|---|---|---|
| | S4 | S5 | S7 | S9 | S14 | S17 | |
| Decision | | | √ | | | | It is suitable for dealing with diverging stakeholder concerns, evaluating technological alter-natives and uncovering relationships between decisions to be made |
| Enterprise | | | | √ | | | It represents the business processes of the target system at architectural level |
| Information | | | | √ | | | It shows the reflection on information models based on the local and/or international terminologies |
| Computational | | | | √ | | | It represents the functional aggregation of the systems components and services |
| Engineering | | | | √ | | | It describes the system infrastructure and mechanisms supporting distribution, in other words, how the system is deployed |
| Technology | | | | √ | | | It shows the architectural model to be implemented |
| Detail | | | | | √ | | It shows information about individual decisions |
| Relationship | | | | | √ | | It shows the relationship between architectural design decisions and their current state in a particular moment in time |
| Chronology | | | | | √ | | It presents all versions of an architectural decision |
| Stakeholder involvement | | | | | √ | | It shows stakeholders responsibilities in the decision-making process |
| Variability | | | | | | √ | It represents the variability in reference architectures |

architecture description and specified from simple protocols for interaction to more complicated requirements regarding information flow.

We also looked for **types of architectural decisions** (e.g., architectural patterns, styles, and technologies) which approaches considered. However, three studies (S4, S7, and S15) only mentioned the possibility of using them, without in fact using them. S4 mentioned the architectural styles client–server, Model-View-Controller, layered, event-driven, and blackboard. S7 mentioned the SOA architectural style, while S15 mentioned the Jade and AGLOBE patterns.

As described along with this section, we can observe that three common elements (architectural view, viewpoint, and model kind) proposed by ISO/IEC/IEEE 42010 are recurrent in the state of the art, but the other six elements are not widely treated in the approaches. **Finding 4:** *The existing approaches do not consider important elements proposed by ISO/IEC/IEEE 42010 that could describe reference architectures suitably.*

The next section examines the description of four reference architectures, also analyzing which elements of ISO/IEC/IEEE 42010 they considered in their descriptions.

## Analysis of four successful reference architectures

**AUTOSAR** (https://www.autosar.org) is a well-known reference architecture for the automotive sector and has brought several significant benefits related to standardization, interoperability facilitation, knowledge reuse, and improvement in the communication among interested parties (e.g., vehicle manufacturers, suppliers, and other companies

**Table 7 Model kinds addressed by the approaches for describing reference architectures.**

| Model Kind | Studies | | | | | | | | | Description |
|---|---|---|---|---|---|---|---|---|---|---|
| | S3 | S4 | S5 | S6 | S7 | S8 | S14 | S17 | S18 | |
| Activity diagram | | ✓ | ✓ | | | ✓ | | | | It shows sequence and conditions for coordinating lower-level behaviors, rather than which classifiers own those behaviors |
| Requirements diagram | | | | | | | ✓ | | | It shows sets of requirements and their relations |
| Parametric diagram | | | | | | | ✓ | | | It enables integration between the design and analysis models. It does this by binding the parameters of the analysis equations that are defined for each analysis to the properties of the subject of the analysis |
| State machine diagram | | ✓ | | | | | ✓ | | | It models the discrete behavior through finite state transitions. Also, it expresses the behavior of a part of the system, state machines can also be used to express the usage protocol of part of a system |
| Use case diagram | | ✓ | ✓ | | ✓ | ✓ | ✓ | | ✓ | It describes a set of actions (use cases) that some system or systems (subject) should or can perform in collaboration with one or more external users of the system (actors) |
| Component diagram | | ✓ | ✓ | | ✓ | ✓ | | | | It shows components and dependencies between them. This type of diagrams is used for component-based development (CBD) and to describe systems with SOA |
| Package diagram | | ✓ | ✓ | | | ✓ | | | | It shows packages and relationships between the packages |
| Workflow diagram | ✓ | | ✓ | | | | | | | It is a visual representation of a business process, usually done through a flowchart. Therefore, it provides a graphical overview of the business process |
| Sequence diagram | ✓ | ✓ | | | | ✓ | | | | It focuses on the message interchange between lifelines (objects) |
| Collaboration diagram | | ✓ | ✓ | | | | | | | It shows objects in a system cooperating with each other to produce some behavior of the system |
| Class diagram | | ✓ | ✓ | | ✓ | | | | | It shows the structure of the designed system at the level of classes and interfaces, shows their features, constraints, and relationships - associations, generalizations, and dependencies. |
| Feature model | | | | ✓ | | | | | | It is a compact representation of all the products of the Software Product Line (SPL) |
| Internal block diagram | | | | | | | | ✓ | | It is a static structural diagram owned by a particular Block that shows its encapsulated structural contents: Parts, Properties, Connectors, Ports, and Interfaces |

from the electronics, semiconductors, and software industry). This architecture was established in 2002 and is currently maintained by the core partners of manufacturers, such as BMW Group, PSA Group, Ford, Toyota, Volkswagen, and Bosch. To maintain the 18-year life cycle, AUTOSAR adopts an update policy with release and version control of its documentation. Similarly, **ARC-IT** (Architecture Reference for Cooperative and Intelligent Transportation) (https://local.iteris.com/arc-it/index.html) was designed by the US Department of Traffic in 1996. After 24 years, this architecture has 13 versions

**Table 8** Stakeholders addressed by the approaches for describing reference architectures.

| Stakeholders | Studies | | | | | |
|---|---|---|---|---|---|---|
| | S4 | S8 | S11 | S14 | S15 | S17 |
| Software architects | | ✓ | ✓ | ✓ | ✓ | |
| System designers | | | | ✓ | ✓ | |
| Leaders | | | ✓ | | | |
| Project managers | ✓ | ✓ | ✓ | | ✓ | ✓ |
| Developers | | ✓ | ✓ | ✓ | | |
| Domain experts | | | | | ✓ | |
| Business-persons | | ✓ | | | | |
| Customers | ✓ | ✓ | | | | ✓ |
| System users | ✓ | ✓ | | | | |
| Engineers | | ✓ | ✓ | | | ✓ |

with updates in the communication standards among intelligent vehicles and refinement in their description. Another reference architecture analyzed is for Industry 4.0. **IIRA** (Industrial Internet Reference Architecture) (https://www.iiconsortium.org/IIRA-1.7.htm) enables architects of industrial internet of things systems to design systems based on a common framework and concepts. The architecture team maintains a living document that is updated to reflect learnings from real-world projects and the latest technologies. **AXMEDIS** (Architecture for Automating Production of Cross Media Context) (http://www.axmedis.org/com/) is also a sustainable reference architecture. In its 15-year life cycle, the AXMEDIS team provided three big releases with updates in standards to content management of partners, such as BBC and HP, and refinement of their description with new viewpoints.

Following, we depict our analysis of how these architectures were described.

### *Architectural description of the four reference architectures*

We examined the documents of the first and last versions of each architecture to measure them regarding $M_{3.1}$ to $M_{3.7}$, which are the metrics specific to RQ3 (previously listed in Table 1) and also $M_{2.1}$ to $M_{2.9}$ that are related to the adherence of the architecture description to ISO/IEC/IEEE 42010. Table 9 summarizes the results of the analysis of these four architectures using these metrics.

These documents were accessed according to how each architecture was disseminated. Some reference architectures offer the description available through websites and pdf documents, while others provide the description in file sets according to architectural modules and even databases. It is interesting to note that the ADL used in each architecture was identified only when we reviewed the latest available documentation because initially some architectures were informally described. For instance, in the middle of the 1990s when ARC-IT was first published, it was informally described. In that time, the first events in the area of software architecture had started and, therefore, the culture of using ADL was not a consensus. We also observe that the adherence to formal and semi-formal ADL was gradual in these architectures.

**Table 9** Summary of the architectural description of AUTOSAR, ARC-IT, IIRA, and AXMEDIS.

| AUTOSAR | Reference Architectures | | | |
|---|---|---|---|---|
| | ARC-IT | IIRA | AXMEDIS | |
| **Domain** | Automotive | Transportation | Industry 4.0 | Media transmission and management |
| M$_{3.1}$ - **Year of establishment** | 2002 | 1996 | 2015 | 2005 |
| **Last analyzed version** | 4.4.0 | 8.3 | 1.9 | 4.5 |
| M$_{3.2}$ - **Pages (first version)** | 2,638 pages 48 files | 1,568 pages 8 files | 100 pages 1 file | 432 pages 4 files |
| M$_{3.2}$ - **Pages (last version)** | 22,271 pages 220 files | 5,204 pages 25 files | 365 pages 4 files | 1,295 pages 13 files |
| M$_{3.3}$ - **Dissemination** | - Website<br>- Documents (.pdf; .zip; .exe)<br>- Models<br>- Meta-models | - Website<br>- Documents (.pdf; .zip; .jpg)<br>- Database | - Website<br>- Documents (.pdf)<br>- White papers<br>- Technical reports | - Website<br>- Documents (.pdf, .iso)<br>- Videos (.wmv)<br>- Player (.mpeg-4) |
| M$_{3.4}$ - **Life cycle (years)** | 17 | 23 | 4 | 14 |
| M$_{3.5}$ - **Number of Releases** | 10 | 13 | 3 | 3 |
| M$_{3.6}$ - **ISO 42010 Adherence Level** | | | | |
| M$_{2.1}$ - Views | Application, Runtime, System Services, Hardware, and Micro-controller | Enterprise, Functional, Physical, and Communications | Business, Usage, Functional, and Implementation views | Simplified view, and Technical view |
| M$_{2.2}$ - Viewpoints | Specification viewpoints (such as use-case, logical, deployment, and implementation) | Enterprise, Functional, Physical, and Communications (diagrams, tables, and databases) | Business, Usage, Functional, and Implementation viewpoints | Not adhered |
| M$_{2.3}$ - Models | Standards divide into Platforms described in UML diagrams, models, and meta-models | "Services Packages" divided into Physical diagrams described in UML, and tables | Component, Sequence, and State Diagrams | Not adhered |

| | Reference Architectures | | | |
|---|---|---|---|---|
| **AUTOSAR** | **ARC-IT** | **IIRA** | **AXMEDIS** | |
| $M_{2.4}$ - Stakeholders | Class of Partners (Core, Premium, Development, and Associate), and Attendees | Class of stakeholders divided according to their role in "Services Packages" | Parties (agent, human or automated) | Not adhered |
| $M_{2.5}$ - Concerns types | Motivation and Goals, Reuse, Quality Attributes, and Safety | Mission, Quality attributes, and Risks | Safety, Security, and Interoperability | Interoperability, and Scalability |
| $M_{2.6}$ - Architectural Decisions (AD) | Architectural patterns, models, and meta- models | Recorded decisions using informal and semi-formal notations | Architectural patterns, Functional maps, and Implementation maps | Workflows, Communication and distribution channels |
| $M_{2.7}$ - Rationale decisions | Constraints and Trade-offs represented asmeta-model | Alternatives and Trade-offs represented in a semi-formal way | Constraints and Trade-offs represented in a semi-formal way | Decisions described in an informal way |
| $M_{2.8}$ - ADL | Informal and Semi-formal (UML) | Informal and Semi-formal (UML) | Informal and Semi-Formal (UML) | Informal and Semi-formal (UML) |
| $M_{2.9}$ - Types of AD | Standards, Software Interface, Communications protocols, Hardware Interface | Layered Style, Communications Profiles, and Correspondence Rules of the domain. | Layered Databus Architecture, Interfaces, Patterns, and Standards | Communications channels, and Editorial Formats |
| **$M_{3.7}$ - Description approaches** | | | | |
| Process | Classic Platform designed using the experience of existing standards | Architecture defined using the experience of existing systems | Architecture defined using the experience of existing systems | Architecture defined using existing technologies |
| Method | Sharing architectural knowledge by Basic Partners (BMW, Bosch, Chrysler, and VW) | Creation and usage of architectural knowledge management by DOT/USA | Sharing architectural knowledge by Industrial Internet Consortium Architecture Task Group | Sharing architectural knowledge by AXMEDIS Consortium |
| Architectural style | Blend of Layered, Modules, and Component-and-Connector Styles | Layered-Style, and Component-and-Connector | Layered-style | Component-and-Connector, and Client-server |
| Document template | Documents follow a design/methodology established by AUTOSAR Release Management Team | Documents follow a design established by DOT/USA Architectural Team | Templates provide by Industrial Internet Consortium | Template established by the European Commissions in IST FP6 |

While new releases have included refinement/extension, this has resulted in a significant increase in the amount of documentation. For instance, AUTOSAR presented in its first version a description with 2,638 pages distributed in 48 files, as seen in Table 9. The current version (4.4.0) has 22,271 pages organized into 220 files, disseminated through a website and documents in pdf, zip, and exe. A good practice adopted by the AUTOSAR architecture team is the description of the change history in each document. This section (referred to as "Document Change History") in each document details information, such as the release date, version, change manager, brief description of the change, and if new standards were adopted or standards were changed. Similarly, the detailed description of ARC-IT has ensured a life cycle that has been sustained over 23 years. Even when it was first proposed in 1996 as a set of standards defining basic services for transportation systems, its description, spread over 1,568 pages, detailed information, such as functional entities, communication services, cost analysis, implementation strategy, and parameters. Currently, version 8.3 features the description on 5,204 pages. Furthermore, the architecture team maintains a database with all data flows, physical, functional components, and communication protocols to facilitate the dissemination and adoption of the architecture.

Analyzing the content of these architectures, it was also found that their description, as it is updated and refined, allows the architecture to be aligned with the state of the art, i.e., the current knowledge of the application domain. For example, version 7.0 of ARC-IT described 22 communications profiles that were developed following closely the naming practices of the OSI (Open Systems Interconnection) Model (*International Organization for Standardization, 2020*). In its latest version, ARC-IT describes 25 communication profiles. In particular, DSRC-UDP (Vehicle-to-Vehicle/Infrastructure using UDP) (*USA, 2019*) is one of these profiles that describe a set of standards applicable to broadcast, frequent, medium latency, and vehicle-to-vehicle and vehicle-to-infrastructure communications using the User Datagram Protocol (UDP). The architecture description details that this communication profile dropped to support the IEE 802 MAC (*IEEE, 2020*) in Data Link Layer and update the communication standard in Presentation Layer, replacing the standard ISO ASN.1 DER to ISO.ASN.1 UPER, which was introduced in 2015.

The dissemination of the description of reference architectures is also a contributing factor to their sustainability. Even with a small life cycle of four years, the dissemination of IIRA across the industry has allowed rapid adoption by the partners. These partners (https://www.iiconsortium.org/IIRA-1.7.htm/) provide feedback as the architecture is instantiated and assist in updating the description. For instance, the latest version of IIRA introduces a refinement of viewpoints and describes more appropriately key crosscutting concerns and their associated system characteristics, such as safety and security. Furthermore, this latest version introduces the Layered Databus Architecture Pattern (*Industrial Internet Consortium, 2020*), a common architecture across IoT systems in multiple industries, offering benefits as (*Industrial Internet Consortium, 2020*): (i) fast device-to-device integration; (ii) automatic data and application discovery; (iii) scalable integration; and (iv) hierarchical subsystem isolation, enabling the development of complex system designs. This same scenario is observed in AXMEDIS (http://www.axmedis.org/com/index.php?option=com_content&task=view& id=80&Itemid=84). In addition to the

website and documents, the AXMEDIS architecture team provides videos (wmv) and players (mpeg-4) with adjustments to standards and laws in the architecture description for the digital content management domain.

We also look at the description of these four architectures from the perspective of the adherence to the ISO/IEC/IEEE 42010 standard. Looking at the first documentation of these architectures, we find that they were initially described at a high-level of abstraction. For instance, the first ARC-IT document for the "Standards Development Plan" (https://local.iteris.com/arc-it/documents/sdp/sdp.pdf), released in 1996, describes a general process to assist standards development and suggests beneficial actions to support and encourage Intelligent Transportation Systems (ITS). Besides that, the document describes specific and potential standards needs for ITS, but this description is at a high-level of abstraction, presenting only some diagrams and other information described in informal language. However, throughout updates, this description has been refined and currently, in its latest version (https://local.iteris.com/arc-it/html/viewpoints/viewpoints.html), the architecture team has organized the standards into groups called profiles, which are represented in viewpoints. Each profile is described detailing the related physical objects, source and destination information flow, data flow, and the required protocols. The same is also observed in AUTOSAR and IIRA, which in their latest versions have descriptions that adhere to all metrics related to ISO/IEC/IEEE 42010 adherence. Here, it is important to note that adherence to ISO/IEC/IEEE 42010 was gradual to AUTOSAR, according to versions released after 2011 when such standard was established. IIRA, established in 2015, has already presented in its first documents descriptions that follow this standard. In the case of AXMEDIS, the architectural description has not yet adhered to all metrics. One of the reasons may be because there was only one update after 2011.

### Good practices for describing reference architectures

As a result of the analysis of the description of these four sustainable reference architectures, it is possible to outline some practices that could be contributing to the sustainability of these architectures:

- **Adherence level to standards**: To effectively serve as a guide for the development, standardization, and evolution of a collection of systems, it is recommended that the description of reference architectures follow known standards. In this way, it is possible for these architectures to communicate in a reliable way the knowledge they contain, considering that a reference architecture involves a huge amount of concerns, stakeholders, and domain experts. The four architectures analyzed presented good adherence of their description to ISO/IEC/IEEE 42010. Other standards exist in the literature (*DeBaud, Flege & Knauber, 1998*; *Dobrica & Niemelä, 2008*; *Cruz et al., 2011*) and can be also used as an alternative.

- **"Living" document**: Keeping documentation updated is another practice observed in good examples of sustainable reference architectures. In particular, the architecture team of these architectures provides the description according to new understandings or refinements that arise over the application domain. These new documents are soon made available (even when a new version of the architecture has not still been released)

because the content of such documents becomes important to keep users, partners, and stakeholders aligned with the state of the art.

- **Summary with change history**: A good practice adopted by the analyzed reference architectures, specially AUTOSAR and ARC-IT, is to present at the beginning of each of their document a summary of the change history. It may contain information, such as major changes from one version to another, the inclusion or exclusion of some view/viewpoint, the adoption of new terminologies, and corrections that may have been necessary for that new version.

- **Availability of a repository with original documents**: Documents that compose the description may be renamed or even merged with other documents. Besides, the terminology may change as understanding of the application domain advances. To avoid misunderstanding in the architectural description, a good practice is to keep a repository containing all original (and/or most important) documents.

- **Organization of the documentation**: Another good practice adopted by these architectures is the facility to find specific information or document. For example, the description of ARC-IT is divided into "Service Packs" where 142 modules are detailed. When the description of each module is opened, information from other views regarding this module, such as enterprise, functional, goals and objectives, needs and requirements, and standards, are also presented in a sub-menu. AUTOSAR also adopts a similar practice.

It is also observed that these good practices for describing reference architectures are directly associated with a significant amount of documentation to be managed (i.e., prepared, updated, and understood for a proper use). Such management is not a trivial, cheap task, requiring a considerable amount of financial and organizational resources and efforts. Hence, the support of large communities and/or consortia that involve various partners (e.g., companies, research centers, universities, and/or others interested) becomes fundamental to the long-term existence of these architectures.

The next section provides an overall discussion encompassing both perspectives (state of the art and state of the practice) concerning the description of reference architectures.

# DISCUSSIONS

Comparing the state of the art (contained in the answer to RQ1 and RQ2, 'Approaches to reference architecture description', 'Analysis of four successful reference architectures' respectively) and the state of the practice (RQ3, 'Analysis of four successful reference architectures'), we observe a considerable misalignment among them. It is worth highlighting that the state of the practice was collected based on four successful reference architectures and, therefore, they could serve as good examples when targeting the sustainability of these architectures. If a random set of reference architectures is considered, including those that have not survived along the time, results could have been different.

Regarding the state of the art, the 19 existing approaches presented different solutions (from processes to ADL) to support the description of reference architectures. In summary, we found that: (i) few approaches consider the international standard for architecture

description (the ISO/IEC/IEEE 42010); (ii) most approaches enable the description of reference architectures in a higher level of abstraction; and (iii) most approaches were not applied and/or validated in real-world scenarios or even in industry. In this sense, we identify important research perspectives that should be still explored:

- **Need for adherence to standards:** The adherence to international standard ISO/IEC/IEEE 42010 is important considering that this standard dictates what is relevant to be considered in architectural description. In general, approaches were not fully adherent to this standard. Additionally, taking into account the few studies that considered the ISO/IEC/IEEE 42010's elements, these elements were described in a high level of abstraction. In this scenario, *approaches that were already proposed or the new ones should systematically incorporate ISO/IEC/IEEE 42010's elements.* As a result, the adoption of these approaches could promote: (i) standardization of reference architectures descriptions; (ii) better understanding of the descriptions; and (iii) improvement in the communication among stakeholders.

- **Need for detailed approaches:** The description of reference architectures is not a trivial activity, because it encompasses different elements that are not sometimes easy to capture. At the same time, such description and associated updates are not cheap tasks, hence requiring the support of companies, governments, or other entities (preferably forming a consortium of partners interested in it). Approaches also need to be detailed with suitable tasks/activities for architecture descriptions. Hence, *approaches should be detailed enough, indicating not only what to do but also how to do, besides the artifacts to be created as well as steps to manage them.*

- **Availability of supporting tools:** The architectural description of a given software system is already naturally a complex, error-prone, and costly task, similar to the description of reference architectures, when manually performed or performed without appropriate tools. For the while, approaches have not given attention to providing associated supporting tools. These tools could automate activities, easing the representation of such architectures. Therefore, *the availability of tools to specifically describe, control, and also instantiate reference architectures is necessary, providing support to different, diverse stakeholders and companies interested in the architectures.*

- **Description of reference architectures of current software-intensive systems:** The size and complexity of software-intensive systems have increased, resulting in what has been referred to as ultra-large systems, systems-of-systems, cyber-physical systems, and others that sometimes present dynamic architectures. In this scenario that involves different partners and even competitors in a target project, reference architectures become even more important. However, approaches did not adequately address both dynamism and interoperability. *The description of such large-scale, dynamic reference architectures should receive special attention together with a change in the mindset of practitioners and researchers regarding the processes to design and evolve them.*

- **Improvement in the collaboration between academia and industry:** Among the studies found in our SMS, only four were proposed and/or validated by industry. In this sense, academia may be conducting research that has not focused on the real

industry needs. Then, it is further necessary to conduct new research to understand the difficulties of the industry to describe reference architectures, together with a wider investigation about the way industry has represented its architectures, as initially presented in 'Analysis of four successful reference architectures'. Besides that, joint research collaboration between academia and industry occurs in different ways depending on the cultural, political, and economic aspects of each country or region. This scenario also requires changes in how to develop research collaboration on reference architectures, including their description. Hence, for the future, *the research on reference architectures description should encompass a proper collaboration between academia and industry, matching real industry needs with a wider evaluation of approaches through real-world reference architectures.*

- **Reference architectures and interoperability:** Despite reference architectures have already contributed to promoting the interoperability among modules and systems implemented following the architectures (*Avgeriou, 2003*; *Valle, Garcés & Nakagawa, 2019*), the studies found in the literature have not explicitly provided means (e.g., model kinds, mechanisms, or others) to describe the interoperability in reference architectures. Therefore, *new approaches to model interoperability in reference architectures and means to deal with such interoperability when instantiating the architectures are necessary.*

Reference architectures themselves need to evolve together with the target application domain that often also continually evolves. Software systems of that domain also evolve according to constantly changing stakeholders' requirements, business rules, technologies, and others, generating new knowledge that can be used as feedback to evolve the reference architecture. Hence, the reference architecture descriptions should be built in such a way that facilitates changes and evolution and, as a consequence, the sustainability of reference architectures over the years. Successful reference architectures as those four analyzed in this work have already provided a way of how they have managed their description and all associated documentation. However, they are isolated cases in the sense each one has addressed the documentation in a way that better works. Some good practices for describing reference architectures could be extracted from investigations like those presented in 'Analysis of four successful reference architectures', but they cannot be considered a generic solution that could work in any architecture. Therefore, more research from a closer collaboration of academia and industry should be conducted to propose a more standardized, generic solution to describe sustainable reference architectures.

## Threats to validity

To minimize biases of our study, we present below the potential threats to validity and actions that we performed to mitigate them:

- **Missing of important studies:** Studies that proposed approaches to describe reference architectures may not have been considered in our analysis. To mitigate this threat, we systematically followed the SMS protocol, besides adopting six databases considering as the most relevant sources in the software engineering area. We also carried out a manual

search using Google Scholar, conferences and journals of the area, technical reports, and book chapters.

- **Data extraction:** During data extraction, we created a data extraction form to fill and save all answers from each study. However, not all the information were obvious to be extracted from the studies and, then, some information had to be interpreted; for instance, the elements of ISO/IEC/IEEE 42010 considered by each approach. In addition, in the event of a disagreement among reviewers, discussions were conducted.

- **Relevance of studies:** The amount and relevance of the studies selected in our SMS may be considered as a threat to validity to the generalization of the results. To minimize this threat, we systematically followed the SMS protocol to select relevant studies together with the entire involvement of all authors of this work.

## FINAL REMARKS

Reference architectures have been increasingly acknowledged for their capability to aggregate the knowledge in various critical, complex domains and support the development and evolution of software-intensive systems in those domains. Hence, an adequate description of these architecture becomes of utmost importance to effectively promote such knowledge reuse and dissemination. Such description becomes even more important in the current scenario where most reference architectures have not survived after their first publication.

This work drew a wide panorama of the means used to describe reference architectures and was built from both the research perspective (which depicted the state of the art) and the practical perspective (which brought scenarios of real-world, successful reference architectures). As a main result, we observe there is a mismatch between the state of the art and the state of the practice. Hence, it is clear the need for developing more integrated research collaboration between academia and industry. While academic research could become aligned to the real-world needs regarding reference architecture description, the industry could benefit from scientific methods already being explored by the software architecture research community.

We also pointed out future research lines highlighting the need for new approaches for reference architectures description that consider important issues, including: (ii) *size and complexity* of reference architectures that involve diverse stakeholders from different partners, segments, and interests; (iii) need to adequately represent *interoperability* in reference architectures in a scenario where large software systems (also referred to as Systems-of-Systems, large-scale systems, cyber-physical systems, and others) have sometimes resulted from the interoperability of diverse constituent systems; and (iv) need to adequately represent *dynamism* in reference architectures, as current software systems have increasingly presented dynamic architectures. More importantly, these approaches must provide the means to assure the sustainability of reference architectures, i.e., the architectural description of these architectures must be developed and organized to primordially facilitate its update and maintenance together with a reduction of required resources and effort.

### Funding

This work is supported by the National Council for Scientific and Technological Development - CNPq (Grants: 312634/2018-8) and the São Paulo Research Foundation - FAPESP (Grants: 2015/24144-7, 2017/06195-9, 2018/07437-9, and 2017/22107-2), and the Beatriz Galindo programme. The funders had no role in study design, data collection and analysis, decision to publish, or preparation of the manuscript.

### Grant Disclosures

The following grant information was disclosed by the authors:
National Council for Scientific and Technological Development - CNPq: 312634/2018-8.
São Paulo Research Foundation - FAPESP: 2015/24144-7, 2017/06195-9, 2018/07437-9, 2017/22107-2.
Beatriz Galindo Programme.

### Competing Interests

The authors declare there are no competing interests.

### Author Contributions

- Pedro Henrique Dias Valle conceived and designed the experiments, performed the experiments, analyzed the data, prepared figures and/or tables, authored or reviewed drafts of the paper, and approved the final draft.
- Lina Garcés and Silverio Martínez-Fernández conceived and designed the experiments, performed the experiments, analyzed the data, authored or reviewed drafts of the paper, and approved the final draft.
- Tiago Volpato performed the experiments, analyzed the data, prepared figures and/or tables, authored or reviewed drafts of the paper, and approved the final draft.
- Elisa Yumi Nakagawa conceived and designed the experiments, analyzed the data, prepared figures and/or tables, authored or reviewed drafts of the paper, and approved the final draft.

### Data Availability

This is a literature review and did not generate raw data.

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
