# Peer review of "Towards suitable description of reference architectures"

_PeerJ Computer Science, doi:10.7717/peerj-cs.392_

## Round 0.1 · original submission · Major Revisions

Both reviewers were quite positive about your submission but suggested resubmission after some revisions, which you can find in their reviews provided below.

·

Basic reporting

The paper is an SLR on approaches for the description of reference architectures. The topic is interesting both for the academic and industrial world. The work is well motivated and well presented. I really enjoyed reading the paper.

Experimental design

The study design is well described.

It may be improved considering the following observations.

The authors consider 4 reference architecture examples which are considered successful. How have they been chosen? For example, the authors cite Volpato et al, 2017 and mention that from the 20 analyzed reference architectures, 12 present no evidence, but it seams that other 8 do present evidence. Did you choose the 4 successful examples from these 8 reference architectures? Based on which criteria? Which are the criteria to consider them successful?

The authors mention the AUTOSAR example with 220 files and 22271 pages of documentation. This implies a significant financial, organizational, besides the development effort. Please comment this aspect related to its success in Section 4.4. The same observation holds for all the 4 examples. In addition, reading, understanding and using such an amount of documentation is not simple. This observation is corelated with the second research perspective in Section 5.

Please insert the exact query you used for each of the search database because they may differ from one to another. For example, you searched the indicated keywords in the whole document or in the title and abstract or in the metadata, etc. Having the exact query allows the replication of the experiment.

Table 2: The most recent approach mentioned in this table is from 2015. Even if the study was conducted in 2018, there are no approaches between 2015 and 2018. Can you please comment this aspect? In addition, I noticed also a gap between 1998 and 2003 with no available approaches.

Line 533: what do you mean by “good examples”? Successful? Well documented?

Validity of the findings

The results are well described and the findings clearly outlined.

Additional comments

Section 5: Improvement in the collaboration between academia and industry: this aspect should be discussed also from the point of with of the cultural and financial aspects worldwide. There are countries in which this collaboration is very strong, in the same there are countries where such collaborations are not supported by cultural and financial rules. Also, the authors of this paper belong to the academic world. Is industry not interested in such an SRL?

Reviewer 2 ·

Basic reporting

The literature review is from the first half of 2018. I would advise to check for more recent papers. Given the structured approach of the paper, any additions should be easy to include.
The quality of English is very acceptable. A check by a native speaker might remove any remaining issues, such as in line 16/17 or 241.

Experimental design

The paper proposes to do a literature search and compare the results with four reference architectures from practice.
For comparison, some sort of reference is required. Such a reference can be developed from the literature, the practice viewed or both. Or it can be found externally. In this case the authors selected an ISO/IEC standard. I would expect an explanation for this choice. Also, it is a bit surprising to use a nine year old standard as the basis for analysis in a (as the authors themselves state) rapidly developing but still immature area.
The selection of reference frameworks from practice appears to have been based on success. This can be made more explicit, and also the definition od ‘success’ is missing (but can be inferred).

Validity of the findings

no comments

Additional comments

Nice work!

---

## Round 0.2 · accepted · Accept

Good work! The referees were both happy with your revision and I am glad to inform you that your paper has been accepted for publication in PeerJ Computer Science.

·

Basic reporting

No comment.

Experimental design

No comment.

Validity of the findings

No comment.

Additional comments

The authors have addressed all the reviewer comments and they have improved the paper based on these comments. I really appreciated the new version of the paper, as well as the motivations given by the authors. Therefore, I suggest the acceptance of the paper for publication.

Reviewer 2 ·

Basic reporting

No comment
This is a re-review. I am satisfied by the way the comments of the reviewers have been addressed.

Experimental design

No comment

Validity of the findings

No comment

Additional comments

No comment